# The Potential Role of Complement System in the Progression of Ovarian Clear Cell Carcinoma Inferred from the Gene Ontology-Based Immunofunctionome Analysis

**DOI:** 10.3390/ijms21082824

**Published:** 2020-04-17

**Authors:** Kuo-Min Su, Tzu-Wei Lin, Li-Chun Liu, Yi-Pin Yang, Mong-Lien Wang, Ping-Hsing Tsai, Peng-Hui Wang, Mu-Hsien Yu, Chia-Ming Chang, Cheng-Chang Chang

**Affiliations:** 1Department of Obstetrics and Gynecology, Tri-service General Hospital, National Defense Medical Center, Taipei 114, Taiwan; aeolusfield@hotmail.com (K.-M.S.); lvita.tw@gmail.com (L.-C.L.); hsienhui@ms15.hinet.net (M.-H.Y.); 2Graduate Institute of Medical Sciences, National Defense Medical Center, Taipei 114, Taiwan; 3Department of Medical Research, Taipei Veterans General Hospital, Taipei 112, Taiwan; backyard0826@gmail.com (T.-W.L.); molly0103@gmail.com (Y.-P.Y.); monglien@gmail.com (M.-L.W.); figatsai@gmail.com (P.-H.T.); 4Division of Obstetrics and Gynecology, Tri-Service General Hospital Songshan Branch, National Defense Medical Center, Taipei 105, Taiwan; 5School of Medicine, National Yang-Ming University, Taipei 112, Taiwan; phwang@vghtpe.gov.tw; 6Department of Obstetrics and Gynecology, Taipei Veterans General Hospital, Taipei 112, Taiwan; 7Department of Medical Research, China Medical University Hospital, Taichung 404, Taiwan

**Keywords:** ovarian clear cell carcinoma (OCCC), gene ontology (GO), immunological function, machine learning, complement system

## Abstract

Ovarian clear cell carcinoma (OCCC) is the second most common epithelial ovarian carcinoma (EOC). It is refractory to chemotherapy with a worse prognosis after the preliminary optimal debulking operation, such that the treatment of OCCC remains a challenge. OCCC is believed to evolve from endometriosis, a chronic immune/inflammation-related disease, so that immunotherapy may be a potential alternative treatment. Here, gene set-based analysis was used to investigate the immunofunctionomes of OCCC in early and advanced stages. Quantified biological functions defined by 5917 Gene Ontology (GO) terms downloaded from the Gene Expression Omnibus (GEO) database were used. DNA microarray gene expression profiles were used to convert 85 OCCCs and 136 normal controls into to the functionome. Relevant offspring were as extracted and the immunofunctionomes were rebuilt at different stages by machine learning. Several dysregulated pathogenic functions were found to coexist in the immunopathogenesis of early and advanced OCCC, wherein the complement-activation-alternative-pathway may be the headmost dysfunctional immunological pathway in duality for carcinogenesis at all OCCC stages. Several immunological genes involved in the complement system had dual influences on patients’ survival, and immunohistochemistrical analysis implied the higher expression of C3a receptor (C3aR) and C5a receptor (C5aR) levels in OCCC than in controls.

## 1. Introduction

Epithelial ovarian carcinoma (EOC) is the fifth most common cause of cancer-related death in women and the most lethal gynecologic malignancy [1]. EOC consists of different histological subtypes, including high-grade serous, clear cell, endometrioid, mucinous, and low-grade serous [2]. Ovarian clear cell carcinoma (OCCC) belongs to type I EOC with clinical, histopathological, and genetic features that are distinct from the other types of ovarian cancer [3]. There is also a distinct difference in the incidence of OCCC among different ethnic populations. In the United States, the incidence rates of OCCC were 4.8% in Caucasians, 3.1% in African Americans, and 11.1% in Asians [4]. In Asia, the incidence rate of OCCC in Japan has increased, and is now over 25% overall, reaching 40% in certain populations [4]. A similar tendency of OCCC incidence rates has been observed in Taiwan [5]. The reasons for the differences in incidence around the world are not known [6]. Gynecological oncologists generally use the Federation of Gynecology and Obstetrics (FIGO) system to classify OCCC into four stages based on disease progression [5]. FIGO staging has been extensively used to evaluate patient survival or patient response to treatment in the past, as well as in current clinical studies. In scaled randomized trials and statistical studies, OCCC was more likely to be found at an early stage (FIGO stage I, around 66.4%) [7], where, conventionally, comprehensive surgical staging via an optimal debulking operation is performed as an initial treatment. Once OCCC progresses beyond this first stage, it is pathologically classified as high-risk (i.e., advanced stages (FIGO stage III, or even stage IV)), where adjuvant chemotherapy is recommended, even if the stage is IA. However, the advanced stages of OCCC are associated with persistent chemoresistance and poor prognosis compared with serous or endometrioid carcinomas, especially in the Asian population, both in terms of progression-free survival (PFS) and overall survival (OS) [8].

Many studies are currently underway to investigate this dilemma. Sampson first reported an association between OCCC and endometriosis in 1925, finding an increased risk of EOC in women with endometriosis, particularly for the clear cell and endometrioid subtype histologies [9,10,11]. The risk of OCCC was found to be predominantly elevated among patients with endometrioma of the ovary (relative risk = 12.4) [10]. It is believed that a subset of OCCC may be developed from ovarian endometriosis via a K-ras mutation, serving as a trigger for carcinogenesis processes [11,12]. Additionally, immunological dysfunction has been proposed as a critical factor involved in endometriosis, and thus OCCC, as endometriosis is a chronic, heterogeneous immunological, inflammatory, and estrogen-dependent disease [13]. To highlight the immunopathophysiology of endometriosis, innate immune mechanisms—including neutrophils, macrophages, natural killer (NK) cells, and dendritic cells (DCs), adaptive immune mechanisms regulated by T lymphocytes (T cells), and antibodies produced by B lymphocytes (B cells), peripheral surrounding immune microenvironment, immune-angiogenesis axis, and immune-endocrine interactions—are crucial facilitators of therapeutic targeting [14]. Thus, OCCC is thought to be derived from endometriosis-associated ovarian cancer (EAOC) and is inseparably correlated to EAOC including endometrioid ovarian carcinoma (EC) and OCCC [3,9,10,11]. As such, we hypothesize that the dysregulated immunological functions in OCCC results in the transformation of normal ovarian tissue (Figure 1).

Tumor mutational burden (TMB) is a predictive biomarker used for evaluating sensitivity to immunotherapy. Ovarian cancer seems to have an acceptable response due to its moderate levels of TMB [15,16,17]. In the past decades, several candidate molecular-targeted drugs, combination chemotherapies, targeted therapies, and check-point-inhibitor-related immunotherapies, including anti-PD-L1 (programmed death-ligand 1)/PD-1 (programmed death 1) single-agent therapy, have been used in clinical trials with dominant preliminary effectiveness [18,19,20,21,22]. As immunotherapy is just one of multiple possible treatments of cancer, the research and development of more effective and powerful immune-related therapeutic approaches, dependent on a thorough understanding of the immunological functions in OCCC, are now being studied. Recent genome-wide studies helped gynecological oncologists better understand the general molecular pathways of OCCC. However, the dysregulated immunological functions of OCCC at different FIGO stages have not yet been measured or entirely quantified within tumors and the tumor environment.

To elucidate the mechanisms and immunological functions of the different OCCC stages, we conducted a whole-genome integrative investigation by analyzing the immunological functions of OCCC detected by microarrays via methods based on differentially expressed genes (DEGs). Subsequently, a functionome-pattern of a gene set regularity model was built, including gene set regularity (GSR) indices of the global functions, to investigate the dysregulated immune-related functions involved in OCCC. Quantifying the immunological functions defined by the Gene Ontology (GO) gene sets allowed us to confirm a significant immunological deterioration during the progression of OCCC between the early (FIGO stage I and II) and advanced stages (FIGO stage III and IV). As such, this analysis and the results presented herein may be conducive to the amelioration and advancement of immunotherapy for OCCC.

## 2. Results

### 2.1. DNA Microarray Gene Expression Datasets for OCCC and Gene Sets Definition

We used the integrative method of gene ontology-based analysis to investigate the immunological function of OCCC. A total of 85 OCCC samples were collected. DNA microarray gene expression datasets were downloaded from the National Center for Biotechnology Information (NCBI) Gene Expression Omnibus (GEO) database, including 23 samples of stage I, 4 samples of stage II, 12 samples of stage III, 5 samples of stage IV, and 41 samples of unconfirmed stages. Numbers of samples and statistics of gene set regularity indices calculated for all the analyzed GO terms and only the immune-relevant GO terms of OCCC stage groups compared with controls are displayed in Table 1. We also collected 136 normal ovarian samples as a control group. The detailed information of collected samples is available in Appendix A. The sample data was collected from 38 datasets containing six different DNA microarray platforms without any missing data (detailed information available in Appendix A). The 5917 GO gene set definitions for annotating the functionome were downloaded from the Molecular Signatures Database (MSigDB) with the version “c5.all.v6.2.symbols.gmt”.

### 2.2. Comparison of Functionomes between the Four OCCC Stage Groups and Normal Controls

The workflow of this study is shown in Figure 2, and a detailed procedure of the computation algorithm is described in the Materials and Methods. First, based on the expression orderings of the gene elements in each gene set, we converted the extracted gene expression profiles of the gene elements from the ordinal data to the quantified 5917 GO terms. We then computed the quantified functions between OCCC and the normal ovarian state, such as the gene set regularity (GSR) indices that ranges from 0 to 1, where 0 indicates the most dysregulated state of a function, i.e., oppositely ordered gene set regularities between OCCC cases and the normal state, while 1 indicates that the regularity in a gene set remains the same between the case and the most common gene expression orderings in the normal controls. Next, we checked the functional regularity patterns and the informativeness of the genome-wide functionome and evaluated the informativeness of the functionome, consisting of the 1454 GSR indices classified and predicted by way of a support vector machine (SVM), a supervised machine learning algorithm. Meanwhile, the functionomes, consisting of 5917 GO gene set defined functions, were reconstructed for each sample. The statistics of the functionomes are listed in Table 1. The differences in the GSR indices between each of the four OCCC stage groups and the normal controls were statistically significant (*p* < 0.05), indicating that the functions were generally dysregulated in the OCCC groups compared with the normal controls.

### 2.3. Reestablishment of Means and Histograms of GSR Indices for Immunofunctionomes and Comparison of the Functionomes to Determine the Relationship between OCCC Stages

There are many immune-related GO terms, which made it difficult to extract all the immune-related GO terms from the GO database directly and precisely. To gather comprehensively associated GO terms and discard irrelevant data, we used two progenitor GO terms, “immune system process” (GO:0002376) and “inflammatory response” (GO:0006954), with 333 offspring extracted from the functionome to create a histogram of the immunofunctionome for comparison between OCCC and normal ovarian tissue samples. Our results revealed that OCCC is initially distinguished from the normal ovarian tissue groups in two different distributions (Figure 3A). We calculated the means of the GSR indices corrected by the averages of the control groups, which decreased stepwise from the early stages (FIGO stage I and II; Figure 3B) to the advanced stages (FIGO stage III and IV; Figure 3C), indicating the steady deterioration of functional regulation with disease progression. Quantifying the regulation of immunological functions by surveying the average of the total GSR indices among each immunofunctionome, with subsequent corrections based on the control groups, the numerical values of the corrected GSR indices for the total, early, and advanced stages were 0.7374, 0.7465, and 0.7309, respectively.

### 2.4. Calculating and Rearranging the Accurate Functional Regulation Patterns of the Early and Advanced OCCC Stages Using Machine Learning

Based on the characterization of the immunofunctionome at different the OCCC stages and in normal ovarian tissue samples, we used machine learning to analyze two progenitors GO terms with 333 descendants from the functionome. According the resulting correlation, we identified 37 dysregulated immune-related GO terms in the early OCCC stages (FIGO stage I and II) and 20 dysregulated immune-related GO terms in the advanced OCCC stages (FIGO stage III and IV). The entire list is obtainable in Appendix A. Here we used cluster weight index (CWI) calculated and ranked by machine learning to quantify and evaluate the importance of each GO cluster in the pathogenesis of OCCC. CWI was defined as the ratio of that cluster weight divided by the sum weight of total clusters to measure the weight and to represent the relevance of each cluster in the GO tree. The higher CWI indicates the higher relevance and higher importance. Ten overlapping immune-related GO terms in both the early and advanced stages were identified(Figure 4), as follows, arranged by correlation from high to low relevance: “regulation of hemopoiesis”(GO:1903706), “regulation of leukocyte migration”(GO:0002685), “regulation of alpha-beta t cell activation”(GO:0046634), “T cell proliferation”(GO:0042098), “antigen processing and presentation”(GO:0019882), “natural killer cell activation” (GO:0030101), “leukocyte homeostasis” (GO:0001776), “spleen development” (GO:0048536), “tolerance induction” (GO:0002507), and “granulocyte differentiation” (GO:0030851).

### 2.5. Twenty-Two Commonly Dysregulated GO Terms Are the Most Meaningful Dysfunctional Immunological Pathways in OCCC Progression

Next, the essential dysregulated functions involved in the progression of disease via malignant transformation from the early stages to advanced stages were extracted by secession of the analytic procedures, including exploratory factor analysis (EFA) and ranking analysis, and filters to elucidate the immune-related pathogenesis of OCCC carcinogenesis. As a result, 156 immune-related dysfunctional pathways were found in the early stages, 178 immune-related dysfunctional pathways in the advanced stages, and 163 immune-related dysfunctional pathways found in all the total stages (FIGO stage I to IV). The complete list is acquirable in Appendix A. We selected the 30 most highly related pathways, in order of relevance from 1st to 30th, from each group and performed cross-comparison to select the 22 immune-related dysfunctional pathways common to each subgroup (Figure 5). Among these dysfunctional pathways, “complement activation alternative pathway” (GO:0006957), “innate immune response in mucosa” (GO:0002227), “natural killer cell activation” (GO:0030101), “negative regulation of leukocyte migration” (GO:0002686), “positive regulation of macrophage activation” (GO:0043032), “regulation of macrophage chemotaxis” (GO:0010758), “natural killer cell mediated immunity” (GO:0002228), “regulation of neutrophil migration”(GO:1902622), “regulation of macrophage cytokine production” (GO:0010935), and “leukocyte degranulation” (GO:0043299) were related to innate immunity, while “regulation of b cell receptor signaling pathway” (GO:0050855), “positive regulation of immunoglobulin secretion” (GO:0051024), “positive regulation of immunoglobulin production” (GO:0002639), “positive regulation of t helper 1 type immune response” (GO:0002827), “humoral immune response” (GO:0006959), “antigen processing and presentation of exogenous peptide antigen via MHC class I” (GO:0042590), “positive regulation of B cell proliferation” (GO:0030890), “regulation of activated T cell proliferation” (GO:0046006), “regulation of isotype switching” (GO:0045191), “negative regulation of cd4 positive alpha beta t cell activation” (GO:2000515), “regulation of t helper 1 type immune response” (GO:0002825), and “negative regulation of lymphocyte mediated immunity” (GO:0002707) were related to adaptive immunity. Notably, the dysfunctional pathway “complement activation alternative pathway” (GO: 0006957) ranked first in all the OCCC stages (both early and advanced) in comparison to the control group.

### 2.6. Distinct Genes Involved in the Key Components of the Dysregulated Immunological Functions Expressed during OCCC Progression

Based on the ranking of “complement activation alternative pathway” (GO:0006957) in the immune-related dysfunctional pathway of OCCC, we used an immune-related gene list, the innate DB [23], to search for immunological genes filtering the differentially expressed genes (DEGs) in the key elements of the dysregulated immunological functions involved in the progression of OCCC, particularly with regards to the GO term “complement activation alternative pathway” (GO:0006957). As a result, we identified 11 immune-related DEGs in all the OCCC stages, including C8G, C8A, CFH, VSIG4, C9, CFP, C8B, C7, C3, CFHR5, and C5, involved in the activation of the alternative complement pathway. The full DEGs list of “complement activation alternative pathway” (GO:0006957) was procurable in Appendix A. We then characterized the role of these immunological genes in the progression of EAOC including EC and OCCC using the Kaplan–Meier plotter (http://kmplot.com/analysis/index.php?p=service&cancer=ovar), a useful online platform and database created by Gyorffy et al. [24]. This tool was used to calculate the prognostic value of the expression levels of all microarray-quantified genes against the survival information of 1816 ovarian cancer patients downloaded from the GEO (https://www.ncbi.nlm.nih.gov/gds) and TCGA (http://cancergenome.nih.gov) datasets, including GSE14764, GSE15622, GSE26712, GSE30161, GSE32062, GSE51373, GSE63885, GSE65986, GSE9891, GSE23603, GSE3149, and TCGA. Quality control and normalization were performed with all the possible cut-off values between the lower and upper quartiles computed to determine the best performing threshold, which was subsequently used as the cutoff. As a result, only the probes present on all three Affymetrix platforms (Affymetrix HG-U133A, HG-U133A 2.0, and HG-U133 Plus 2.0 microarrays) were retained (*n* = 22,277). We calculated the progression-free survival (PFS) and overall survival (OS) in 62 endometriosis-associated ovarian cancer (EAOC) FIGO stage I/II/III/IV patients with chemotherapy of platin and taxane. This allowed us to investigate further the relationship between OCCC patients’ survival and the expression levels of the DEGs with regards to the potential dysregulated immunological functions. Interestingly, seven complement-related immunological genes (CFP, VSIG4, C9, C8B, C7, C3, and C5) relevant to progression-free survival (PFS) and overall survival (OS) were identified via the correlation between OCCC survival and the DEGs involved in the activation of the complement alternative pathway. High expression levels of three immunological genes (CFP, C9, and C5) were found to be significantly correlated with good prognosis and survival; on the contrary, the high expression levels of another four immunological genes (VSIG4, C8B, C7, and C3) were significantly correlated with poor prognosis and survival. The hazard ratios of PFS and OS of these immunological genes are shown in Appendix A. These results suggest that the 7 genes in the complement activation alternative pathway play an ergastic immunological role in the promotion of OCCC disease progression. Moreover, their prognostic value in OCCC and the complement system may have dual effects on OCCC, and activation of alternative complement pathway may act as both a promoter and an inhibitor of OCCC.

### 2.7. The Immune-related Genes of the Complement System Have Influence on Progression of OCCC

Next, we searched the GEO repositories to recognize suitable datasets for the analysis and found that GSE65986 dataset contained OCCC subtype with the raw data extracted and corrected. All clinical data and gene expression were integrated into a PostgreSQL relational database. We then compared and calculated the 7 DEGs (CFP, VSIG4, C9, C8B, C7, C3, and C5) in the complement activation alternative pathway selected from the above step and to investigate the relationship between OCCC patients’ survival and the expression levels of the DEGs with regards to the potential core dysregulated immunological functions via the Mann–Whitney test and the receiver operating characteristic test in the R statistical environment (www.r-project.org) using Bioconductor libraries (www.bioconductor.org) and the Affymetrix platforms (Affymetrix HG-U133A, HG-U133A 2.0, and HG-U133 Plus 2.0 microarrays) followed by a second normalization to set the average expression of the 22,277 identical probes [24]. The inferred results of the progression-free survival (PFS) in 25 OCCC FIGO stage I/II/III/IV patients with chemotherapy of platin and taxane revealed that C3 had synchronous poor effects on PFS with statistical significance (Figure 6A). Furthermore, based on the previous survival analysis, we utilized the seven immunological markers (CFP, VSIG4, C9, C8B, C7, C3, and C5) related to the complement activation alternative pathway and STRING database (https://string-db.org) to build a functional protein–protein-interaction network (Figure 6B). As members of immune/inflammasome pathway related genes, the seven proteins showed intensive interactions and regulatory cross effects. This interactive network supported the proposed involvement and potential role of complement system in OCCC malignant progression and C5 revealed stronger and closer relationship than the other markers.

### 2.8. Immunohistochemistrical Analysis of Anti-C3aR and Anti-C5aR Expression between OCCC and Normal Ovarian Tissues

Since C3 and C5, the complement-related markers, were inferred from previous analysis to have influences on OCCC progression and to evaluate the clinical significance of the identified immune or inflammasome-related markers in OCCC transformation by way of primary validation is essential. As the lead meaningful dysfunctional GO term found in this research is “complement activation alternative pathway (GO: 0006957)”, which means any process involved in the “activation” of any of the steps of the alternative pathway of the complement cascade which allows for the direct killing of the microbes and the regulation of other immune process (https://www.ebi.ac.uk/QuickGO/term/GO:0006957). The previous research had revealed that the complement activation, an important innate immune response, was regarded with the help of elevated anaphylatoxins—complement component 3a (C3a) and complement component 5a (C5a) and the mechanistic role of the binding of C3a/C5a to their respective receptors (C3aR and C5aR) had been proved to take a place in the progression of inflammation or continued immune response [25]. In the process of activation of the alternative pathway of complement, in fact, the most representative roles related to the inflammatory response should be C3a and C5a, the intermediate products of the alternative pathway of complement. C3a is a small peptide of C3 from initial hydrolysis reaction and via C3 convertase disintegration; however, C3b is extremely unstable under normal conditions and would immediately form C3 convertase (C3bBb) and C5 convertase (C3bBbC3b) with other components during complement-related reactions or be used for reactions such as opsonization. C5a is a peptide product from C5 dissociated by C5 convertase while C5b is quickly utilized as the part of the membrane-attack-complex (MAC). C3a and C5a can be said to be the peptide mediators of inflammation for proinflammatory signaling and anaphylatoxin effects during the process of activation of the alternative pathway of complement, while C3a receptors and C5a receptors distributed on endothelial cells and immune-related cells in the tumor microenvironment could combine with C3a and C5a to activate G proteins for triggering subsequent reactions (https://www.ncbi.nlm.nih.gov/books/NBK27100/). To explore the role of immune/inflammasome-related and inflammatory responses in the transformation and progression of OCCC. To assess the specific meaning clinically of the recognized immune-related genes of activation of alternative pathway of complement system involved in transformation of OCCC, we had gathered a cohort of clinical samples (OCCC, *n* = 12; normal control group, *n* = 12) and immunostained them with anti-C3aR and anti-C5aR antibody separately. Therefore, we used immunohistochemistrical analysis of anti-C3aR and anti-C5aR antibody among OCCC and control groups for measuring clinical significance of C3 and C5 indirectly due to their higher protein expression compared to other immune-related genes (The human protein atlas: https://www.proteinatlas.org/). Theoretically, C3a and C5a could represent C3 and C5 symbolically, because C3a and C5a are small polypeptides released from the cleavage of their precursors C3 and C5, and both of these molecules are of a high degree of similarity, and furthermore, C3a receptor (C3aR) and C5a receptor (C5aR) recognize C3a and C5a specifically to exert their functions [25,26]. We found more increased C3aR and C5aR expression level in OCCC samples than in normal samples (Figure 7A,B). Quantification of C3aR and C5aR levels, with help of Graphpad prism software, in OCCC samples showed a higher mean value of C3aR and C5aR expression than in the control group with statistical significance (Figure 7C,D). These results suggest that dysfunctional activation of the alternative pathway of complement system collaborates with other subordinate deregulated immunological genes or pathways and plays dual roles in promoting and demoting OCCC progression. These results offer initial clinical evidence sustaining the involvement of complement system, especially the alternative pathway, in the malignant transformation and progression of OCCC.

## 3. Discussion

In previous studies, molecular biology and genetic research using recent microarray experiments has shown that OCCC is distinct from other EOCs and has a unique genetic profile, characterized by frequent ARID1A (50%) and PIK3CA (40%) mutations, MET amplification, and rare p53, BRCA1, and BRCA2 mutations; however, p53 and PTEN mutations are also frequently observed in ovarian cancers in general [27]. The high frequency of ARID1A and PIK3CA mutations in OCCC theoretically results in a higher activity of the PI3K–AKT–mTOR pathway, which is believed to be a critical pathway on basis of the genomic characterization of OCCC; on the other hand, phosphoinositide 3-kinase (PI3K)-related pathways are known to be essential in the carcinogenesis of clear cell carcinoma, endometrioid carcinoma, and mucinous carcinoma [28]. However, most of the results have either involved minimal clinical activity or activity that was not superior to traditional treatments [18]. A relationship of the immune system and inflammation has been previously identified between endometriosis, endometriosis-associated ovarian cancer (EAOC), and ovarian clear cell carcinoma (OCCC). As such, this study was aimed to determine the core immunological functions and related gene or dysfunctional pathways in relation to the immune-inflammasome. The immune response is triggered as a result of the immune system being activated by antigens and inflammation in a host. It is part of the complex biological response of body tissues to harmful stimuli, and functions as a protective response involving immune cells, blood vessels, and molecular mediators to eliminate the initial cause of cell injury, eliminate necrotic cells and tissues damaged from the original insult and the inflammatory process, and initiate tissue repair [29]. Inflammation is associated with the secretion of inflammatory cytokines, which lead to the formation of an inflammatory microenvironment, considered to be a characteristic feature of cancers, and, therefore, is a critical modulator of carcinogenesis [30]. Although ovarian carcinoma tumor cells can be recognized and attacked by the immune system initially, specific immune-escape biological responses are needed in the ovarian carcinoma tumor cells and the surrounding microenvironment at the time of the activation of molecular anti-tumor mechanisms [31].

Traditionally, differentially expressed genes (DEGs) detected by microarrays have been utilized to investigate different immunological functions. We established a gene-based set regularity model that could be used to re-establish functionomes, for example the gene set regularity (GSR) indices of the global functions, to subsequently investigate the dysregulated functions involved in the complex disease in comparison to DEGs. Rebuilding the functionome can provide us information on the dysregulation of key functions for complex diseases, such as ovarian carcinomas [32,33,34,35]. We previously integrated microarray gene expression profiles downloaded from publicly available databases to execute several gene set-based analyses. Using these research methods, we confirmed that the dysregulation of the cell cycle in EOCs, including dysregulated oxidoreductase activity, metabolism, hormone activity, inflammatory responses, innate immune response, and cell–cell signaling, played a critical role in the malignant transformation of endometriosis-associated ovarian carcinoma (EAOC), to which OCCC is associated [34].

In this study, we applied a similar method to investigate the differences in terms of the immunofunctionomes of OCCC samples and control samples. The resulting histograms of the GSR indices for the immunofunctionomes of the four OCCC stage groups and the control group were more evidently distinct at the progression of OCCC from the early to advanced stages (Figure 2). To identify the mechanism that results in this trend, we used gene expression datasets downloaded from the GEO database combined with support vector machine (SVM), a controlled set of mathematical instructions of machine learning, to compare OCCC in the early and advanced stages against normal ovarian tissue in order to recognize and classify individual patterns among the four different stage groups using 5917 GO gene set defined functions, 333 immune-related functionomes, and 622 functional immunological pathways. As a result, we found that the most disordered immunological functions in OCCC (sorted by statistical significance) were associated with the activation of the alternative complement pathway, which is coordinated with other dysregulated immunological functions in the development of OCCC.

Although some differences existed with regards to the immune-related GO genes and dysfunctional pathways among the subgroups (early and advanced stages) of OCCC in this study, several dysregulated immune-related functions were found to take place proportionally during OCCC progression via both innate and acquired immunity, including the regulation of hemopoiesis, leukocyte migration and homeostasis, T cell activation and proliferation, antigen processing and presentation, natural killer cell activation, spleen development, tolerance induction, and granulocyte differentiation. These dysregulated immune-related functions were also found to affect the prognosis and survival of OCCC patients. These results were used to confirm that the role of immune-inflammatory responses accounted for a certain proportion of the effects in the OCCC carcinogenesis, and even affected the prognosis and survival rate of OCCC patients. These findings correlate with the results published by several previous studies. In addition, we compared the immunofunctionomes of the early stages (FIGO stages I and II) and advanced stages (FIGO stages III and IV) to determine whether they were different. As a result, the advanced stages (FIGO stages III and IV) were found to have more significant differences between the OCCC samples and normal ovarian tissue, indicating that the biological effects of the immunofunctionomes in the advanced stages (FIGO stages III and IV) of OCCC, relative to the early stages (FIGO stages I and II), play a more critical factor.

We found that 22 major dysfunctional immune-related biological pathways, selected from a list of the top 30 in the early, advanced, and total stage of OCCC using explicit machine learning, were involved in the progression of OCCC, classified according to innate or acquired immunity. Subsequently, 10 immune-related dysfunctional pathways were found to be related to innate immunity, including the complement activation alternative pathway, innate immune response in mucosa, natural killer cell activation and mediated immunity, regulation of neutrophil migration, regulation of macrophage-related positive activation, cytokine production and chemotaxis, leukocyte-related degranulation, and negative regulation of migration. Another 12 immune-related dysfunctional pathways were found to be related to innate immunity containing humoral immune response, antigen processing and presentation of exogenous peptide antigen via MHC class I, regulation of isotype switching, positive regulation of B cell proliferation, regulation of B cell receptor signaling pathway, positive regulation of immunoglobulin secretion and production, (positive) regulation of T helper 1 type immune response, negative regulation of CD4(+) T cell activation, and the negative regulation of lymphocyte-mediated immunity. In short, many mechanisms were found to have a strong influence either in the promotion of disease progression or in the suppression between OCCC and its surrounding environment via both innate and acquired immunity.

Interestingly, we also found that the key role of the complement system, especially with regards to the dysfunctional activation of the alternative pathway (GO:0006957), took place in the transformation from normal ovarian tissue to all stages of OCCC. The role of the complement system in several inflammatory disorders has been previously investigated, as complement, a specific combination of proteins found in the blood and body fluids and on cell surfaces, was discovered over 100 years ago to be a heat-sensitive component of the plasma, enhancing the opsonization of foreign bacteria or pathogens, and has traditionally been described as a “complement” to humoral immunity. The complement system is now perceived as a central constituent of innate immunity, helping various events during inflammation to work efficiently and bridging the innate and adaptive immune responses to safeguard the host against from the invasion of pathogens [36]. There are three major pathways for activating complement system. Antigen-antibody complexes could activate the classical pathway, wherein the binding of pattern-recognizing mannose-binding lectins (MBLs) gives rise to lectin pathway. Any indulgent surfaces or interfaces could result in the activation of the alternative pathway. Finally, all complement activation pathways result in the formation of the C3 convertase complex on the surface of targeted cells, which exerts an effector function and results in the formation of the pore-like membrane attack complex (MAC), leading to cell lysis. The role of the complement system in cancer has been previously discussed in many studies, wherein the complement system has been found to modulate the immune response in the tumor microenvironment and has a dual role in tumorigenesis. In response to cancer cells, the complement system generates the lytic membrane attack complex (MAC) on the targeted cell, working with CD8(+) T cells, mature dendritic cells, M1 macrophages, T help 1 cell, N1 neutrophils, and natural killer cells to elicit an anti-tumor immune response. However, the sub-lytic membrane attack complex (MAC)—also produced by complement system—is an accomplice to cancer, whereby it induces tumor proliferation by impeding tumor apoptosis and coalesce with T regulatory T cells, myeloid-derived suppressor cells (MDSC), M2 macrophages, T help 2 cells, and N2 neutrophils to promote angiogenesis and tumor survival, effectively suppressing the anti-tumor response. Several binary components are also produced during the activation of the complement system, such as C5a, C3a, and C1q, which play both anti-tumorigenic and pro-tumorigenic roles in ovarian and other cancers [37]. In this study, our results suggested that the complement system, especially with regards to the activation of the alternative pathway, collaborates with other subordinate dysregulated immunological functional genes or pathways to exert a dual role in the promotion and inhibition of OCCC progression; therefore, as well as having a prognostic value in OCCC. In the following, we discuss the individual functions of the seven genes of the alternative pathway of the complement system among different stages of OCCC.

### 3.1. CFP/Complement Factor Properdin

This gene encodes a plasma glycoprotein with mild to moderate immune-active appearance in ovarian tissue that positively regulates the alternative complement pathway of the innate immune system [38]. Down-regulation of CFP to activation of alternative pathway of complement system to affect progression of OCCC was mentioned in Appendix A. Binding of CFP with the C3- and C5-convertase enzyme complexes in a feedback loop on many microbial surfaces and apoptotic cells enhances the stabilization of formation of the membrane attack complex that ultimately leads to lysis of the target cell [39]. In this study, CFP was calculated regarding down-regulation to the activation of an alternative pathway of the complement system.

### 3.2. C9/Complement C9

This gene encodes the final component of the complement system with moderate cytoplasmic immunoreactivity was observed in ovarian tissue. It participates in the formation of a complement C5b-9 membrane attack complex (MAC). The MAC assembles on membranes of bacteria or intruders, as well as cancer cells, to form a pore, disrupting the bacterial membrane organization and causing cell damage [38]. MAC-induced cancer cell death is also called complement-dependent cytotoxicity (CDC) [40]. However, several membrane complement regulatory proteins on the cancer cell surface and soluble complement regulators in the cancer microenvironment could collaborate, leading to pathological cancer cell resistance to CDC. Elevated levels of soluble C5b-9 found in the intraabdominal ascitic fluid from ovarian cancer patients suggested complement activation [41].

### 3.3. C5/Complement C5

This gene encodes a component of the complement system with moderate to strong immunoreactivity on ovarian tissue, a part of the innate immune system, that plays an important role in inflammation, host homeostasis, and host defense against pathogens. The encoded preproprotein is proteolytically processed to generate multiple protein products, including the C5 alpha chain, C5 beta chain, C5a anaphylatoxin, and C5b. The C5 protein is comprised of C5 alpha and beta chains, which are linked by a disulfide bridge. Cleavage of the alpha chain by a convertase enzyme results in the formation of the C5a anaphylatoxin, which possesses potent spasmogenic and chemotactic activity, as well as the C5b macromolecular cleavage product, a subunit of a MAC [38]. Effects of regulation on the activation of an alternative pathway of the complement system for the progression of OCCC were mentioned in Appendix A. Aslan et al. recently reported that overexpression of complement C5 in endometriosis [42] and cancer cells may secrete complement proteins such as C3a and C5a, resulting in complement activation in the tumor microenvironment through the PI3K/AKT signaling pathway to increase cell proliferation. Complement activation products, such as C3a and C5a, activate their receptors on cancer cells, that through PI3K/AKT signaling increase cell proliferation [43]. Increased soluble C5b-9 in ascitic fluid from ovarian cancer patients suggested complement-related activation [41]. Selene et al. also reported that C5aR antagonist causes complement inhibition, resulting in blocking tumor outgrowth and ovarian cancer progression [44].

### 3.4. VSIG4/V-set and Immunoglobulin Domain Containing 4

This gene encodes a v-set and immunoglobulin-domain containing protein, V-set and Ig domain-containing 4 (VSIG4), that is structurally related to the B7 family-related macrophage protein with a potential role in cancer due to its capacity to inhibit T-cell activation [45]. This protein is also a receptor for the complement component 3 fragments C3b and iC3b. Alternate splicing results in multiple transcript variants [38]. Byun et al. found that overexpressed levels of VSIG4 in ovarian cancers was noted compared to benign ovarian tumors with association of progression and recurrence of ovarian cancer to predict prognosis [46]. VSIG4 has general effects of up-regulation and the activation of an alternative pathway of the complement system for affecting the progression of OCCC (Appendix A). In our current study, VSIG4 was calculated in relation to the up-regulation and activation of alternative pathways of the complement system.

### 3.5. C8B/Complement C8 Beta Chain

This gene encodes one of the three subunits of the complement component 8 (C8) proteins, with mild to moderate immune-active appearance in ovarian tissue [38]. C8, composed of equimolar amounts of alpha, beta, and gamma subunits and encoded by three separate genes, is a component of MAC, mediating cell lysis and initiating membrane penetration of the complex. This protein mediates the interaction of C8 with the C5b-7 membrane attack complex precursor, and forms a C5b-8 complex, which polymerizes several C9 molecules, forming the cytolytic MAC [47]. General up-regulation of C8B, and the activation of an alternative pathway of the complement system for the progression of OCCC is also noted in Appendix A. Increased levels of C5b-9 complex from the intraabdominal ascitic fluid of patients with advanced ovarian cancer was mentioned previously [41].

### 3.6. C7/Complement C7

This gene encodes a serum glycoprotein with mild to moderate immune-active appearance in ovarian tissue forming MAC together with the complement components C5b, C6, C8, and C9, as part of the terminal complement pathway of the innate immune system. The protein encoded by this gene contains a cholesterol-dependent cytolysin/membrane attack complex/perforin-like (CDC/MAC/PF) domain, and has effects involved in host immunity and bacterial pathogenesis. C7 initiates the formation of MAC by binding the C5b-C6 subcomplex and serving as a membrane anchor to insert into the phospholipid bilayer of the cell membrane [38]. Complement-related activation may lead to elevated soluble C5b-9 in ascitic fluid from ovarian cancer patients [41], and the up-regulation of C7 to the activation of an alternative pathway of the complement system to influence the progression of OCCC is also noted in Appendix A.

### 3.7. C3/Complement C3

Complement component C3 plays a core role in the activation of the complement system. Both classical and alternative complement activation pathways could produce C3 convertase and then lead to its activation. The encoded preproprotein with moderate to strong immunoreactivity on ovarian tissue is proteolytically processed to generate alpha and beta subunits to form the mature protein, and then further processed to generate numerous peptide products [38]. The C3a peptide, modulating inflammation and possessing antimicrobial activity, could contribute to tumor proliferation, invasion and metastasis, and is also shown to be crucial for leukocyte infiltration of the tumors [37], and generalized up-regulation effects and the activation of an alternative pathway of the complement system during the progression of OCCC in our analytic results were mentioned in Appendix A. Cho et al. had found that the complement system could promote tumor growth through activated C3aR and C5aR via the PI3K/AKT pathway, and that the knockdown of complement system in cancer cells reduces cell proliferation and cancer growth via silencing the PI3K or AKT gene in cancer cells, causing the elimination of the effects of C3aR and C5aR stimulation [43,44].

This study has several limitations. Firstly, this operational model has limitations as below: the first limitation is that the GO gene set databases do not collect all human functions yet; the second limitation is the detectability of the GSR model, because this model converts gene expression levels to ordinal data, and the GSR index will remain unchanged and aberrations will be missed if the expression levels do not reach the detection levels; the third limitation is the false positivity arising from the duplicated elements existing in different gene sets; the fourth limitation is the some non-immune/inflammation GO terms are included in the immunofunctionome, leading to the bias of statistics for immunofunctionome, such as the means and SDs; the fifth limitation comes from the heterogenicity of cellular composition in tumor and control samples. The datasets utilized in study are composed of the gene expression profiles from the mixture of immune and tumor cells. So, the differences of GSR indices may arise from the gene expressions of differing sampled cellular compositions and may not exactly reflect a deregulated process. Secondly, due to the small number of OCCC clinical samples, we simplified and classified the four FIGO stages into two groups, early stages (FIGO stage I and II) and advanced stages (FIGO stage III and IV). As there may be more obvious and significant differences in each stage compared to normal ovarian tissue, we believe that future studies may prefer to take the time to obtain more clinical samples and analyze the data thoroughly to obtain more meaningful and differentiated outcomes for the OCCC stages. Thirdly, only two major groups of serous subtypes containing mainly high grade serous ovarian carcinoma (HGSOC) and endometrioid subtypes, including ovarian clear cell carcinoma (OCCC) and endometrioid ovarian carcinoma (EC), were established in the study of Kaplan–Meier plotter [24]. The prognosis and survival of OCCC was calculated with the help of the databases from the results of this study. However, no whole exclusive data from OCCC independently quoted but only endometrioid subtype cited to represent OCCC may lead to a problem of anamorphosis. Endometriosis associated ovarian cancer (EAOC) contains OCCC and EC that seem to be homologous originated from endometriosis and we had proposed the similarity between OCCC and EC in previous study via gene-set based integrative analysis [32,33]. Due to the lack of a complete and independent data base for OCCC, we use a common shared database as a last resort to estimate the corresponding PFS and OS of these DEGs involved in the activation of the alternative pathway of the complement system among OCCC. The process seemed to be quite flawed, but the results obtained in this method could be used to explain the impacts of related genes on the prognosis (PFS and OS) of OCCC via the activation of alternative pathways of the complement system. In the next experimental design, of course, a whole complete database containing only OCCC samples is still needed, and it is also the goal that we would need to work toward in the future. Fourthly, the methodology used in this experiment was to use immunohistochemistrical analysis of anti-C3aR and anti-C5aR expression between OCCC and normal ovarian tissues to indirectly measure the performance of C3a and C5a to prove that the dysfunctional pathway, “complement activation alternative pathway” (GO: 0006957), has a certain role and influence in the process of OCCC transformation and progression. The elevated expression of both anti-C5aR and anti-C3aR via the immunohistochemistrical analysis of OCCC with statistically significance was noted, and this may imply that the complement system has a dual action as a promotor or controller in cancer. To understand the changes in the middle connection of pathogenic changes, it is necessary to obtain intermediate transitional cells during the process of transformation and progression from endometriosis to OCCC. This part is quite difficult in most time, so that research in this area in the world is still insufficient. We conducted this study mainly via methods of using big data extracted from the GO database in dry lab and the results of this study could be utilized as the basis and direction for further experimental verification in the future. However, it needs more biomarkers related to the other collaborated core activation genes of the alternative pathway in the complement system involved in the progression of OCCC, and more clinical samples for the validation of the proposed mechanism and the elimination of extreme value. Continued verification requires large-scale experimental tests and funding for collecting samples and continuing the experiments.

Gene ontology-based immunofunctionome analysis is applied in this study to evaluate the immunological effects of OCCC in comparison to normal ovarian tissue. The results provide several distinct immune-related factors during the disease progression of OCCC. We found that the dysfunctional activation of the alternative pathway of the complement system, with seven significantly influential genes, was a crucial factor that could offer clinical targets and relevant biomarkers for the detection, monitoring, and treatment of OCCC. Based on the results of this study, more experimental verification of the effects of these complement-related genes and proteins on the transformation and progression of OCCC need to be performed in the future. Our findings could also be combined with precision medicine to elevate the effectiveness of related treatments, improve the survival and therapeutic quality of patients, and avoid the occurrence of OCCC in the future.

## 4. Materials and Methods

### 4.1. Computing the GSR Indices and Reconstructing the Functionome and Immunofunctionome

The GSR index was computed from the gene expression profiles by modifying the differential rank conservation (DIRAC) [48] algorithm, which measures the changes of ordering among the gene elements in a gene set between the gene expression profiles of OCCC and the most common gene expression ordering in the normal control samples. The details of the GSR model and the computing procedures are described in our previous study [31,32]. Microarray gene expression profiles for OCCC and normal ovarian samples were downloaded from the GEO database. The corresponding gene expression levels were extracted according to the gene elements in the GO gene set and converted to ordinal data based on their expression levels. The GSR index is the ratio of gene expression ordering in a gene set between the case and the most common gene expression ordering among the normal ovarian tissue samples. The measurement of GSR indices was carried out in R. A functionome is defined as the complete set of biological functions. At present, the definition for comprehensive biological functions is not yet available; as such, we annotated the human functionome by the 5917 GO gene set defined functions. The functionome in this study is defined as the assembly of 5917 GSR indices for each sample. The immunofunctionome was then reconstructed by extracting the offspring from the immune-related ancestor GO terms “immune system process” (GO:0002376) and “inflammatory response” (GO:0006954) from the functionome.

### 4.2. Microarray Dataset Collection

The selection criteria for the microarray gene expression datasets from the GEO database was as follows: (1) the OCCC samples and normal control samples should originate from ovarian tissue; (2) the datasets should provide information about the diagnosis and the stage of OCCC; (3) any gene expression profile in a dataset was discarded if it contained missing data.

### 4.3. Statistical Analysis

The Mann–Whitney U-test was used to test the differences between the major two OCCC stage groups and the controls, and then corrected by multiple hypotheses using the false discovery rate (Benjamini–Hochberg procedure). The *p*-value was set at *p* < 0.05.

### 4.4. Classification and Prediction by Machine Learning

The function “svm” provided by the “kernlab” (version 0.9–27; Comprehensive R Archive Network; https://cran.r-project.org/), an R package for kernel-based machine-learning methods, was used to classify and predict the patterns of the GSR indices. The accuracies of the classification and predictions by SVM were measured by k-fold cross-validation. The results of 10 repeated predictions were used to assess the performance of binary classification with sensitivity, specificity, accuracy and area under curve (AUC). AUC was computed using the R package “pROC” [49]. The performance of multiclass classification was assessed using the 10 repeated prediction accuracies for each of the distinct OCCC stage groups.

### 4.5. Cluster Weight Index

The GO terms were summarized by cluster weight index (CWI). The CWI, ranging from 0 to 1, is an index measuring the ratio of the weight for a given cluster accounting for the pathogenesis by calculating the weighting based on the p values and is measured by summing all the negative logarithms of the p values from all the elements in each cluster.

### 4.6. Set Analysis

All possible logical relations among the dysregulated gene sets of the classified OCCC stage groups were displayed in Venn diagram using the R package “Venn Diagram” (version 1.6.16; Comprehensive R Archive Network; https://cran.r-project.org/). Graphpad prism software (https://www.graphpad.com/scientific-software/prism/) was used to perform paired t-test to quantify and compare the levels of C5aR and C3aR between the OCCC and the normal group.

### 4.7. Clinical Samples

The present study included 12 archived OCCC cases (OCCC, *n* = 12) and 12 controlled cases (normal, *n* = 12). In the cases of ovarian cancer, tissues were collected from women who underwent surgery as their treatment guideline. The patients were diagnosed and treated and had their tissues placed in a bank at the Tri-Service General Hospital, Taipei, Taiwan. All invasive cancers were confirmed by histopathology. The Institutional Review Board of the Tri-Service General Hospital, National Defense Medical Center approved the study (2-107-05-043, approved on 26 October 2018; 2-108-05-091, approved on 20 May 2019). Informed consent was acquired from all patients and control subjects.

## 5. Conclusions

Ovarian clear cell carcinoma (OCCC) is a potentially lethal epithelial ovarian cancer (EOC), secondary to high-grade serous ovarian carcinoma (HGSOC). A characteristic feature of OCCC is its resistance to chemotherapy and poor overall survival rate. In this study, whole-genome integrative sequencing was used to determine the global immunological functions of OCCC at different stages. Via the analysis of the OCCC immunofunctionome, a significant difference was found in terms of the immunological functions between OCCC and normal ovarian tissue samples, with a greater disparity found in the advanced stages of OCCC than in the early stages. From the analysis of the immunofunctionome, 10 immune-related dysregulated GO terms were identified, including terms for the regulation of hemopoiesis, leukocyte migration, leukocyte homeostasis, T cell activation, T cell proliferation, antigen processing and presentation, natural killer cell activation, spleen development, tolerance induction, and granulocyte differentiation. Additionally, 22 immune-related dysfunctional pathways in the OCCC groups were also found, wherein the dysregulated activation of an alternative pathway of the complement system was ranked first in all groups, with a statistical significance.

Furthermore, several complement-related immunological genes involved in the activation of the alternative pathway of the complement cascade were found to have double-faced effects on the progression-free survival (PFS) and overall survival (OS) of OCCC patients with either good or poor prognosis (Figure 8). Immunohistochemistrical analysis of anti-C3aR and anti-C5aR expression between OCCC and normal ovarian tissues insinuated higher expression of C3 and C5 level in OCCC than controls statistically. This illustrates the two-sided nature and dual role of the complement system as a friend or a foe for OCCC. These findings provide a reliable molecular biomarker for the monitoring of OCCC and present potential targets for immunotherapeutic treatment of OCCC.

## Figures and Tables

**Figure 1 ijms-21-02824-f001:**
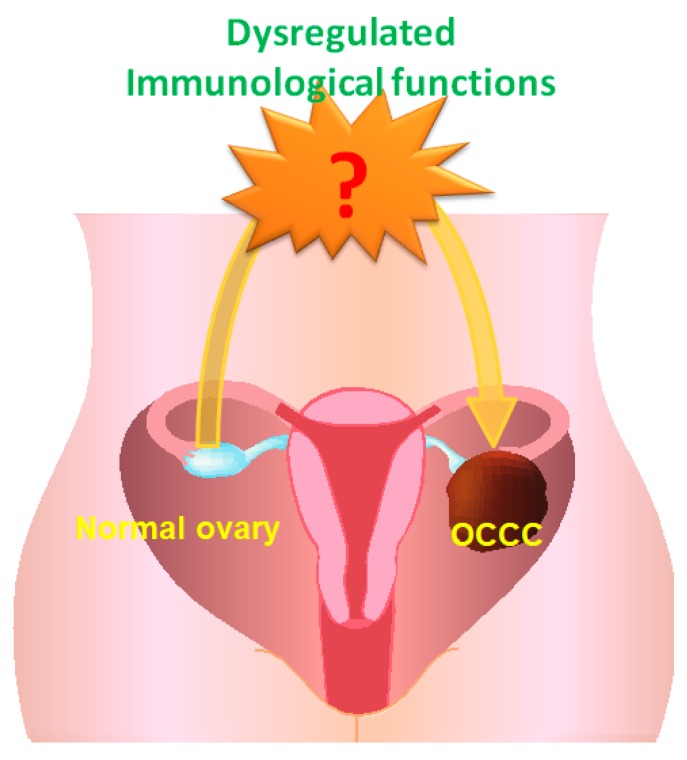
Schematic diagram of proposed hypothesis for dysregulated immunological functions of OCCC transformed from normal ovary.

**Figure 2 ijms-21-02824-f002:**
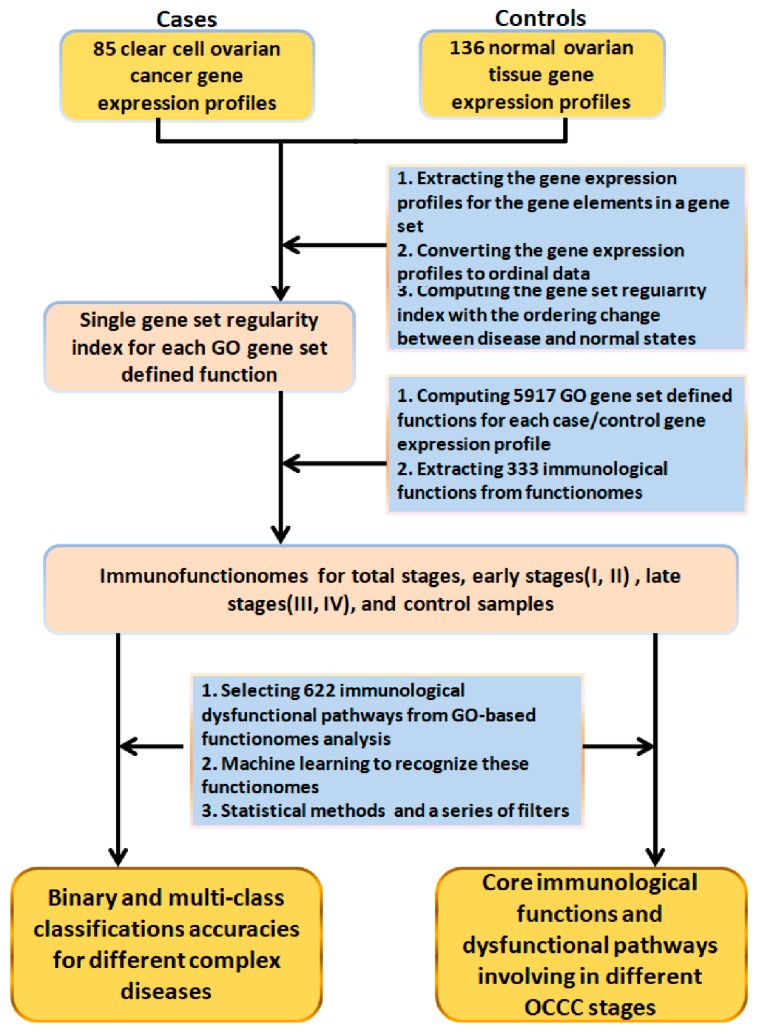
Study workflow. The DNA microarray gene expression datasets of 85 OCCC groups, including the total stages for the general screening, the early and advanced subgroups according to the FIGO system, and 136 normal ovarian controls, were downloaded from a publicly available database. The gene set regularity (GSR) index was calculated by measuring the changes in gene expression ordering of the gene elements in the Gene Ontology (GO) gene set. A functionome consisting of 5917 GO gene sets was reconstructed for each sample. Then, an immunofunctionome consisting of 333 immunological functions was rebuilt by extracting the immune-ancestor GO terms from the functionome for the OCCC and normal control groups. Machine learning using a support vector machine (SVM) was used to recognize and classify the patterns of the functionomes. The essential immunological functions were extracted using statistical analysis and a series of filters.

**Figure 3 ijms-21-02824-f003:**
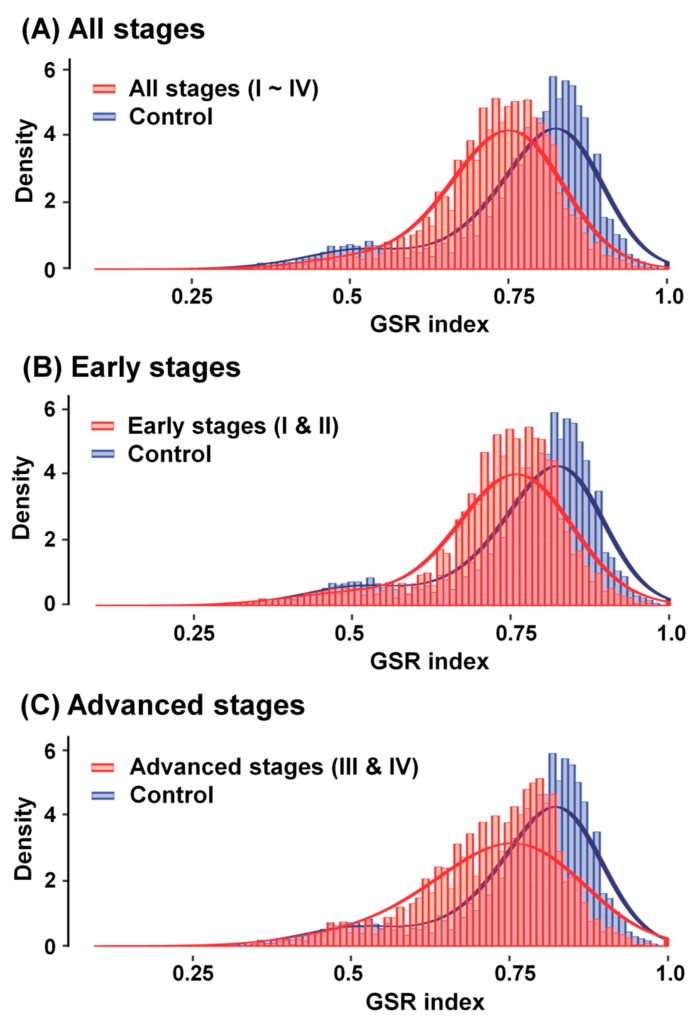
Histograms of the GSR indices for the immunofunctionomes of OCCC and control groups. The normal ovarian tissue group (blue) on the right-hand side of histogram was utilized as control for the distinct OCCC stage groups (red) on the left-hand side with an apparent shift in deviation: (**A**) the total-stages of OCCC: GSR indices: 0.7374, (**B**) the early-stages: GSR indices: 0.7465 and (**C**) the advanced-stages: GSR indices: 0.7309.

**Figure 4 ijms-21-02824-f004:**
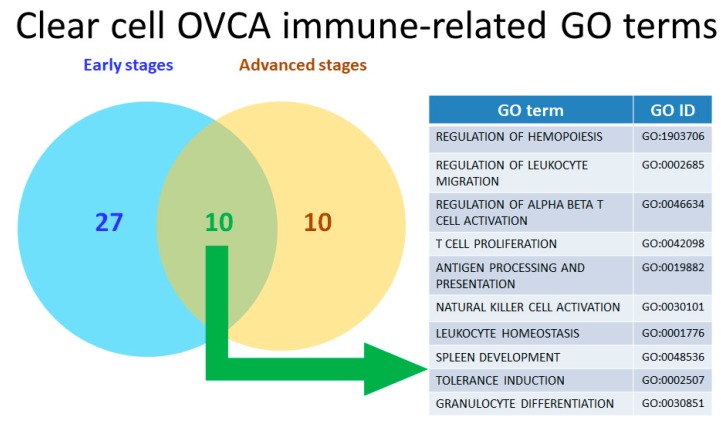
Venn diagram of dysregulated clear cell OVCA (ovarian cancer) immune-related GO terms. The results of the set analysis for the early and advanced OCCC groups with dysregulated immunological functions are displayed on the Venn diagram (left). There were 37 dysregulated immune-related GO terms in the early OCCC stages (FIGO stage I and II) and 20 dysregulated immune-related GO terms in the advanced OCCC stages (FIGO stage III and IV). The 10 common dysregulated GO terms are listed in the table (right).

**Figure 5 ijms-21-02824-f005:**
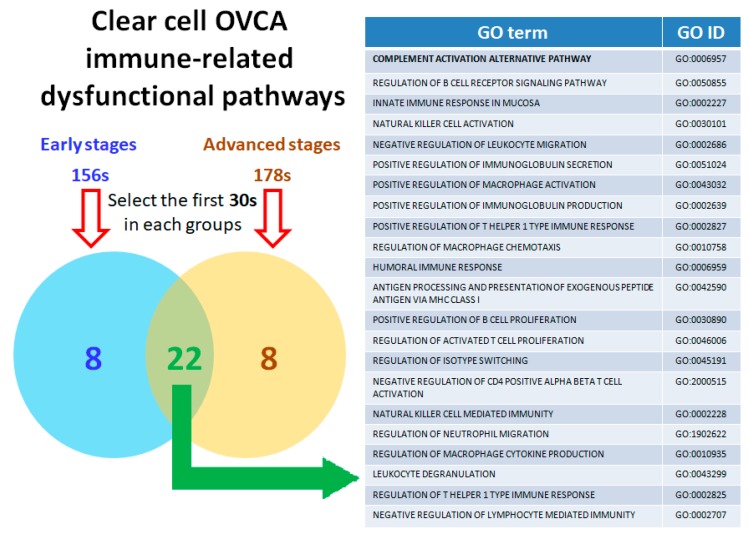
Venn diagram of clear cell OVCA (ovarian cancer) immune-related dysfunctional pathways. The 22 core elements of the immunofunctionome involved in the progression of OCCC from the early to the advanced stage are listed in the table (right). The dysfunctional pathway “complement activation alternative pathway” (GO: 0006957) was ranked first in all stages of OCCC in comparison to the control group.

**Figure 6 ijms-21-02824-f006:**
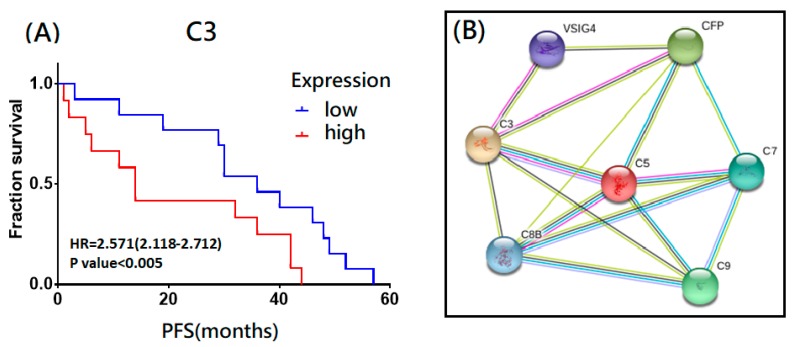
The immune-related markers of the complement system have influence on progression of OCCC. (**A**) The immune-related genes (C3) of the complement system associated with poor survival outcomes (progression-free survival (PFS)) in OCCC. The hazard ratios of the PFS of C3 were 2.571 (2.118-2.712, *p* < 0.005). (**B**) The identified potential involving DEGs were subjected to a protein-protein interaction (PPI) analysis by establishing an interactive network from the STRING database (https://string-db.org) with intensive interactions. The average node degree is 4.29, and the PPI enrichment *p*-value is <0.001, significantly stronger interactions than expected. C5 revealed stronger and closer relationship than the other markers.

**Figure 7 ijms-21-02824-f007:**
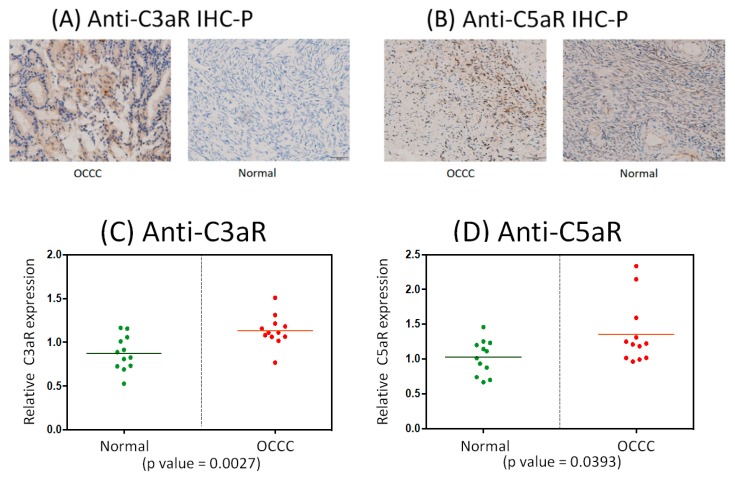
Immunohistochemistrical analysis of clinical samples from patients with OCCC and normal controlled group. (**A**) Clinical samples from patients with OCCC (*n* = 12) and normal group (*n* = 12) were immunostained with anti-C3aR antibody (yellow-brown color). (**B**) Clinical samples from patients with OCCC (*n* = 12) and normal group (*n* = 12) were immunostained with anti-C5aR antibody (in light yellow-brown color). (**C**–**D**) The expression levels of C3aR and C5aR in all clinical samples were quantified and presented in the chart. The mean values of C3aR and C5aR expression in OCCC were higher than those in the normal group.

**Figure 8 ijms-21-02824-f008:**
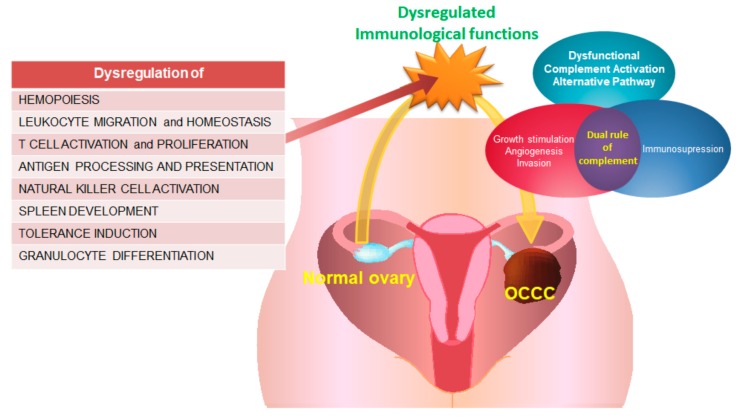
The proposed immunopathological mechanism involved in progression of OCCC.

**Table 1 ijms-21-02824-t001:** Numbers of samples and statistics of gene set regularity indices for OCCC stage groups compared with controls. The table above contains numbers of samples and statistics of gene set regularity indices for all analyzed GO terms and the table below includes numbers of samples and statistics of gene set regularity indices only for the immune-relevant GO terms.

**Group of Stage**	**Sample**	**Control**	**Total**	**Case Mean (SD** **^1^)**	**Control Mean (SD** **^1^)**	***p*-value**
Early stage (stage I and II)	27	136	163	0.7465(0.1114)	0.7745(0.1284)	<0.05
Advanced stage (stage III and IV)	17	136	153	0.7309(0.1176)	0.7744(0.1282)	<0.05
N/A^2^	41	136	177	0.7374(0.1040)	0.7745(0.1286)	<0.05
**Group of Stage**	**Sample**	**Control**	**Total**	**Case Mean (SD** **^1^)**	**Control Mean (SD** **^1^)**	***p*-value**
Early stage (stage I and II)	27	136	163	0.7328(0.1045)	0.7687(0.1239)	<0.05
Advanced stage (stage III and IV)	17	136	153	0.7255(0.1080)	0.7687(0.1239)	<0.05
N/A^2^	41	136	177	0.7269(0.0980)	0.7682(0.1239)	<0.05

^1^ SD, standard deviation; ^2^ N/A, unconfirmed stages.

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
