# Peer review of "The Potential Role of Complement System in the Progression of Ovarian Clear Cell Carcinoma Inferred from the Gene Ontology-Based Immunofunctionome Analysis"

_ijms, 2020, doi:10.3390/ijms21082824_

Round 1

Reviewer 1 Report

The manuscript titled “Gene ontology-based immunofunctionome analysis disclosing the role of complement system in progression of ovarian clear cell carcinoma” describes the analysis of DNA microarray gene expression datasets for OCCC, comparing functionomes between the four OCCC stage groups and normal controls. Authors evaluate the gene set regularity (GSR) indices, showing the dysregulation during the OCCC. Moreover, they identify 37 dysregulated immune-related GO genes in the early OCCC stages  and 20 dysregulated immune-related GO genes in the advanced OCCC stages. Then they analyse immune-related dysfunctional pathways, highlighting that the pathway “complement activation alternative pathway” ranked first in all the OCCC stages. Between these genes they identified 11 immune-related DEGs in all the OCCC stages. Some of them result significant. Authors show an involvement of the activation of the alternative pathway of the complement system in OCCC progression, highlighting the deregulation of some complement system activation components.

The work is interesting, the paper is well written and results are well presented. However, some revisions are necessary.

Revisions:

36 and 284: “2.8. Immunohistochemistrical (!!) analysis of anti-C3aR and anti-C5aR expression between OCCC and normal  ovarian tissues.” These experiments are too much indirect. The expression of a receptor is not always proportionally related to ligand levels. Moreover, C5aR expression had not statistical significance. In my opinion it should be removed.

179: “arranged by correlation from high to low relevance”: In Table S2 the genesets are arranged from high to low P-value and in figure 4 they have the same arrangement. What does “relevance” means in this case? Please check if the order needs to be inverted.

233: “we identified 11 meaningful immune-related DEGs in all the OCCC stages”. Not every DEG in the list have a statistically significant p-value, therefore “meaningful” is not suitable. Otherwise, it should be explained.

243: “This allowed used to”à “This allowed us to”

257 The hazard ratios are also in the figures and they appear hard to read in the text. If you want to highlight values, you could change the font size in the figure.

Discussion: If the activation of the alternative pathway of the complement system took place in the transformation from normal ovarian tissue to all stages of OCCC, some points deserve discussions. In example, VSIG is a potent inhibitor of the alternative complement pathway convertases, how do you explain its upregulation in OCCC? In a same manner, CFP is a positive regulator of the alternate pathway of complement, how do you explain its down regulation in OCCC?

Author Response

Thanks for your insight comments. 

Reviewer 2 Report

Su et al. report in their submitted manuscript about the identification of Gene Ontology-defined processes related to the immune system with disturbed process-specific gene expression orders between normal ovarian tissue and ovarian clear cell carcinoma. They used public available gene expression data from 85 cancers and 136 normal tissues, defined GO processes and adopted the published DIRAC method for the analysis. Additionally they analyzed a few tissue samples for C3aR and C5aR. This paper mainly consists of bioinformatics and system biology methods/results and is rather descriptive and hypothesis-generating. Several points exclude the publication of the manuscript:

Major remarks:

In general the results are hard to interpret, descriptive only and any validation is missing.

  1. The authors used available genome wide expression data from cancer and normal tissue to calculate gene set regularity (GSR) indices for specific GO processes. Thus detected differences between GSR indices are potentially caused by (i) deregulated processes itself or (ii) varying tissue compositions. Depending on the specific sampling strategies in the original studies, generating the gene expression profiles, the tissues have variable fractions of normal (epithelial) cells, immune or tumor cells. Additionally, not all genes of a specific GO process are exclusively active within this process. Thus differences in gene expression ranks for a GO process may originate from differing tissue compositions and may not reflect a deregulated process. Thus, without additional data the results cannot be interpreted.
  2. The authors point to the relation between endometriosis and EOC subtypes to explain their aim to analyze deregulated immune system related functions. However they did not include any endometriosis samples to prove this connection.
  3. The statistics for the GSR indices are calculated for all analyzed GO terms but should include only the immune-relevant GO terms which are the topic of the paper.
  4. The complete manuscript contains false statements. Presented data point to differences in disease states or tissue composition only and no connection to causative changes can be derived. Moreover detected differences in GSR cannot be evaluated properly because any validation is missing.
  5. The gene-specific analysis for the prognostic value of the gene expression (KM-plotter) uses data for endometrioid ovarian cancer and cannot support any data from the GSR analysis for ovarian clear cell carcinoma.
  6. The immunohistochemistry data are limited in the number of cases analyzed. Moreover these data are hard to interpret because not C3/C5 are analyzed but their receptors.

Author Response

Thanks for your constructive comments. Please see the attachment.

Reviewer 3 Report

Su et al. wrote an manuscript “Gene ontology-based immunofunctionome analysis  disclosing the role of complement system in  progression of ovarian clear cell carcinoma, which contained some interesting findings but overall lacks accuracy and sufficient data to support the conclusion that “seven immunological genes (CFP, C9, C5, VSIG4, C8B, C7, and C3) involved in the complement system  had a significant influence on patient survival and immunohistochemistrical analysis implied  higher expression of C3 and C5 level in OCCC than controls.”

The authors have used the gene expression profiles of 85 ovarian clear cell cancers and 136 normal controls from 38 datasets obtained from 6 different microarray platforms. They compared   “functionomes” using gene set regularities  between the four OCCC stage groups and normal controls. Next they identified immune-related GO terms overlapping between low and high stage OCCC and identified  10 immuno-related genes present in both early and late stage OCCC. Next they identified gene sets that are in involved in OCCC progression.  They further investigated 11 genes from the top immune-related dysfunctional pathway (complement activation alternative pathway) and looked at the involvement of these genes in the PFS and OS of endometrioid ovarian cancer  (not OCCC) using the KM plotter. Finally they determined expression of receptors binding to small  cleavage products from the C3 and C5 proteins comparing a small set of normal vs OCCC tissues.

Major comments:

1. It seems that the authors are using the data from the different datasets (38) without correcting for batch effects or normalization. Why not?

2. The authors determine the relation of genes with  PFS and OS using survival data from endometrioid ovarian cancer in KM plotter. To my knowledge no OCCC is in these data. But even if this is a mixed bag of endometrioid, mucinous, and clear cell ovarian cancer results have to be taken with extreme caution, since these ovarian cancer subtypes are not identical.

3. The IHC analyses are not really adding anything. The authors show that C3 and C5 gene expression levels is related to PFS and OS. The authors only indicate differences in protein levels of receptors binding to cleaved peptides from C3 or C5  between normal and OCCC patients, but do not confirm/validate that these protein levels  are related to OS or PFS within OCCC patients.

4. Many of the correlations are mentioned in tables or  figures and in the text, which is not needed.

Minor comments:

Line 200 1stto 30th

Line 229 a immune

Line 258 CFP:  0.23 (0.05-1.01, p=0.034) is not in line with the figure text

Author Response

Thanks for your insight comments. Please see the attachment.

Round 2

Reviewer 2 Report

Su et al. report in their revised manuscript about the identification of Gene Ontology-defined processes related to the immune system with disturbed process-specific gene expression orders between normal ovarian tissue and ovarian clear cell carcinoma. The authors changed parts of their manuscript and discussed critics raised by the reviewer. However, this reviewer still sees several points excluding the publication of the manuscript. In general any validation is missing (see points 1, 4-6), the manuscript still contains false statements (see point 4) and the description of methods is insufficient excluding a proper verification by the reader. Of course authors should state and discuss limitations of their study but if the limitations overweight the validated results authors should think about how to overcome the limitations or how to show the validity of their results. For the submitted paper, the reviewer thinks this will need additional experiments and data analysis – not only a discussion of limitations.

Major remarks:

In general the results are hard to interpret, descriptive only and any validation is missing.

1. The authors used available genome wide expression data from cancer and normal tissue to calculate gene set regularity (GSR) indices for specific GO processes. Thus detected differences between GSR indices are potentially caused by (i) deregulated processes itself or (ii) varying tissue compositions. Depending on the specific sampling strategies in the original studies, generating the gene expression profiles, the tissues have variable fractions of normal (epithelial) cells, immune or tumor cells. Additionally, not all genes of a specific GO process are exclusively active within this process. Thus differences in gene expression ranks for a GO process may originate from differing tissue compositions and may not reflect a deregulated process. Thus, without additional data the results cannot be interpreted.

Response 1:

Thank you for providing us with so many valuable suggestions. Because of the heterogeneity of cellular composition in the OCCC and normal ovarian samples, the results may not exactly reflect the gene expression from immune cells. This bias can be resolved by single cell analysis; however, such data is still not available in the public available database currently. So, we add this situation to the limitation in the Discussion section as the highlighted sentence:

This model has limitations. The first limitation is that the GO gene set databases do not collect all human functions yet. The second limitation is the detectability of the GSR model. Because this model converts gene expression levels to ordinal data, the GSR index will remain unchanged and aberrations will be missed if the expression levels do not reach the detection levels. The third limitation is the false positivity arising from the duplicated elements existing in different gene sets. The fourth limitation is the some non-immune/inflammation GO terms are included in the immunofunctionome. This will lead to the bias of statistics for immunofunctionome, such as the means and SDs. The fifth limitation comes from the heterogenicity of cellular composition in tumor and control samples. The datasets utilized in study are composed of the gene expression profiles from the mixture of immune and tumor cells. So, the differences of GSR indices may arise from the gene expressions of differing sampled cellular compositions and may not exactly reflect a deregulated process.

Added remark (V2): The required analysis must not include single cell analyses but a solid validation of the proposed deregulated immune functions/processes e.g. by gene expression analyses on independent samples or immunohistochemistry.

2. The authors point to the relation between endometriosis and EOC subtypes to explain their aim to analyze deregulated immune system related functions. However they did not include any endometriosis samples to prove this connection.

Response 2:

Thanks sincerely for your truly constructive comment. In the section of introduction and discussion, we had all mentioned and cited past studies, such as " Sampson reported an association between OCCC and endometriosis in 1925, finding an increased risk of EOC in women with endometriosis, particularly for the clear cell and endometrioid subtype histologies"1., and we had also compared the deregulated immune-related functions and inflammasome between endometriosis and EAOC including OCCC and endometrioid ovarian cancer compared to normal ovarian tissue in the previous research (quoted Figure 1. as below)2.,3.. Therefore, the goal of this experiment was extended and focused to

immune/inflammasome-related functions and GO between OCCCs and normal ovarian tissues. The results of this investigative study will provide an informative direction for future researches on immune-related function and inflammasome among endometriosis as well as endometriosis associated ovarian cancer (EAOC) and normal ovarian tissues. It can also be applied to further related experiments involving endometriosis specimens.

Added remark (V2): Either the aim of this work does not include endometriosis  but stage specific analyses of OCCC (most parts related to endometriosis should be excluded) or data from endometriosis patients and an analysis of the complement pathway must be included.

3. The statistics for the GSR indices are calculated for all analyzed GO terms but should include only the immune-relevant GO terms which are the topic of the paper.

Added remark (V2): The required data were added.

4. The complete manuscript contains false statements. Presented data point to differences in disease states or tissue composition only and no connection to causative changes can be derived. Moreover detected differences in GSR cannot be evaluated properly because any validation is missing.

Response 4:

Thanks heartily for your positive comments. All the time from the past, epithelial ovarian cancer (EOC) could be classified as type I EOC including OCCC, endometrioid and other rare ovarian cancer and type II EOC including mainly high-grade serous ovarian cancer (HGSOC). To understand the changes in the middle connection of pathogenic changes, it is necessary to obtain intermediate transitional cells during the process of transformation and progression from endometriosis to OCCC. This part is quite difficult in most time, so that research in this area in the world is still insufficient. We conducted this study mainly via methods of using big data extracted from GO database in dry lab and the results of this study could be utilized as the basis and direction for further experimental verification of EAOC/OCCC in the future. This study used a data-driven and functionome-based analysis methods to infer the immune/inflammasome-related pathological cause of OCCC. However, continued verification requires large-scale experimental tests and funding for collecting samples and making the experiments on going. This is a defect of this study and this defect had been written in the section of "DISCUSSION".

Added remark (V2): It is not absolutely required to prove the contribution to OCCC development and progression but to correctly state that only associations are detected. However, a validation of the proposed deregulated immune functions/processes is needed (see point 1). Otherwise the detected differences may originate form differences in immune cell frequency only. Additionally, the authors claim significant results in the text for some analyses showing p>0.05.

5. The gene-specific analysis for the prognostic value of the gene expression (KM-plotter) uses data for endometrioid ovarian cancer and cannot support any data from the GSR analysis for ovarian clear cell carcinoma.

Response 5:

Thanks sincerely for your constructive comment at first. In our study, the role of these immunological DEGs in the transformation and progression of OCCC was characterized with the help of the Kaplan–Meier plotter (http://kmplot.com/analysis/index.php?p=service&cancer=ovar), a useful online platform and database created by Gyorffy et al. However, only two major groups of serous subtypes that contain high grade serous ovarian carcinoma (HGSOC) and endometrioid subtypes that contain ovarian clear cell carcinoma (OCCC) and endometrioid ovarian carcinoma (EC), were established in the study of Kaplan–Meier plotter (as Table 1 revealed below).4. Endometriosis associated ovarian cancer (EAOC) contains OCCC and EC and we had proposed the similarity between OCCC and EC in previous study via gene-set based integrative analysis (as Figure 1).5. As mentioned in the section of "discussion", due to the lack of a complete and independent data base for OCCC, here we use a common shared database containing both two subtypes of ovarian cancer (OCCC and endometrioid cancer) to estimate the corresponding PFS and OS of these genes (CFP, C9, C5, VSIG4, C8B, C7, and C3) involved in activation of alternative pathway of complement system among OCCC. Although OCCC and EC seem to be homologous originated from endometriosis as type I epithelial ovarian cancer (EOC)6. 7., they are still different in histology. The process seemed to be quite flawed, but the results obtained in this method could be used to explain the impacts of related genes on the prognosis (PFS and OS) of OCCC via complement system. In the after experimental design, of course, a whole complete database containing only OCCC samples is still needed, and it is also the goal that we would need to work hard in the future.

Added remark (V2): It is not correct that the KM-plotter analyses a set of EOC consisting of endometrioid AND clear cell carcinoma but it analyses endometrioid EOC only. Thus the presented data cannot validate the GSR analysis for OCCC.

6. The immunohistochemistry data are limited in the number of cases analyzed. Moreover these data are hard to interpret because not C3/C5 are analyzed but their receptors.

Response 6:

At first, thanks sincerely for your constructive comment. The main meaningful dysfunctional GO term found in this research is "complement activation alternative pathway" (GO: 0006957), which means any process involved in the "activation" of any of the steps of the alternative pathway of the complement cascade which allows for the direct killing of the microbes and the regulation of other immune process (https://www.ebi.ac.uk/QuickGO/term/GO:0006957). Our study mainly explored the role of immune/inflammasome-related and inflammatory responses in the transformation and progression of OCCC. In the process of activation of the alternative pathway of complement, in fact, the most representative roles related to the inflammatory response should be C3a and C5a, the intermediate products of the alternative pathway of complement (as Figure 2 below mentioned).

C3a is a small peptide of C3 from initial hydrolysis reaction and via C3 convertase disintegration, however, C3b is extremely unstable under normal conditions and would immediately form C3 convertase(C3bBb) and C5 convertase(C3bBbC3b) with other components during complement-related reactions or be used for reactions such as opsonization. The second is C5a which was a peptide product from C5 dissociated by C5 convertase while C5b is the part of the membrane-attack-complex (MAC). C3a and C5a can be said to be the peptide mediators of inflammation for proinflammatory signaling and anaphylatoxin effects during the process of activation of the alternative pathway of complement, while C3a receptors and C5a receptors distributed on endothelial cells and immune-related cells in the tumor microenvironment could combine with C3a and C5a to activate G proteins for triggering subsequent reactions (https://www.ncbi.nlm.nih.gov/books/NBK27100/) 8.. As a results, the methodology used in this experiment was to use immunohistochemistrical analysis of anti-C3aR and anti-C5aR expression between OCCC and normal ovarian tissues to indirectly measure the performance of C3a and C5a to prove that the dysfunctional pathway, "complement activation alternative pathway" (GO: 0006957), has a certain role and influence in the process of OCCC transformation and progression. Theoretically, C3a and C5a could represent C3 and C5 symbolically because C3a and C5a are small polypeptides released from cleavage of their precursors C3 and C5 and both of these molecules are of high degree of similarity, and furthermore, C3a receptor(C3aR) and C5a receptor(C5aR) recognize specifically C3a and C5a to exert their functions.

Added remark (V2): It is clear but irrelevant that C3a/C5a measurements can replace C3/C5 measurements but the presence of the receptor does not inform about the activation of the complement pathway. Only if the authors can cite publications showing that the amount of receptor correlates with activation of the complement system an IHC for the receptor may validate their analyses. However the number of cases is still too low and the evaluation should not analyze the general staining but cell type specific intensity (e.g. epithelial cells).

Author Response

Dear reviewer:

Thanks for your insight comments. 

Reviewer 3 Report

The authors have modified the manuscript  according to the comments. Part of the added text is difficult to read and/or contains errors. 

For instance 
line 512: Thus, VSIG levels are associated progression and recurrence of ovarian cancer, including OCCC.

Author Response

Dear reviewer:

Thanks for your insight comments. We reply as below.

Point 1: The authors have modified the manuscript according to the comments. Part of the added text is difficult to read and/or contains errors.

For instance

line 512: Thus, VSIG levels are associated progression and recurrence of ovarian cancer, including OCCC.

Response 1:

Thanks kindly for your positive comments. We had made the whole manuscript polishing retouching and had deleted the incorrect sentence “Thus, VSIG levels are associated progression and recurrence of ovarian cancer, including OCCC”. (line 552)

Round 3

Reviewer 2 Report

Albeit the authors adressed the critical points raised by the reviewer the changes do not substantiate a publication of the manuscript. Specifically the added validation data (GSE 65986) for the proposed prognostic value of C3 expression (Fig. 6) is inaccurate. The authors used PFS data without stratification regarding patients status (progressed or censored). Most patients from GSE65986 (Uehara 2015; PLoS ONE 10(6): e0128066.doi:10.1371/journal.pone.0128066) have censored survival data.